# The Optimization of the Hot Water Extraction of the Polysaccharide-Rich Fraction from *Agaricus bisporus*

**DOI:** 10.3390/molecules29194783

**Published:** 2024-10-09

**Authors:** Aya Samy Ewesys Khalil, Marcin Lukasiewicz

**Affiliations:** 1Department of Food Engineering and Machinery for Food Industry, Faculty of Food Science, Agricultural University in Krakow, al. Mickiewicza 21, 31-120 Krakow, Poland; aya.samy@agr.cu.edu.eg; 2Food Science Department, Faculty of Agriculture, Cairo University, Giza 12613, Egypt

**Keywords:** *Agaricus bisporus*, white button mushroom, extraction, polysaccharides, optimization

## Abstract

The optimization of extraction parameters, including the process time, temperature, and liquid-to-solid ratio, was conducted in order to obtain the polysaccharide-rich fraction from the lyophilized *Agaricus bisporus* fruiting body. The efficiency of extraction for polysaccharides and antioxidant activity was determined by analyzing the extracts for total carbohydrate content, the reducing sugars content, and the antioxidant activity employing DPPH, ABTS, and hydroxyl radical scavenging assays. The results showed that all parameters, except for the extraction time, impacted differently on the extraction efficiency of polysaccharides and antioxidant activity. The highest total carbohydrate content was observed at the longest process time, highest temperature, and a liquid-to-solid ratio of 118 mL/g. To minimize the reducing sugar level, a lower temperature is required, while the highest antioxidant activity requires a moderate temperature and the lowest liquid-to-solid ratio. The optimization of antioxidant activity by means of the DPPH and H_2_O_2_ method failed, which shows that the specific mechanism of polysaccharides as antioxidants needs further investigation. The aqueous extraction method demonstrated to be an efficient and simple approach to recover the potentially bioactive polysaccharide fractions from *Agaricus bisporus* that are also active as antioxidants.

## 1. Introduction

The current era of climate change has prompted researchers and food producers to reevaluate global food production strategies [1,2]. It is imperative to identify new, eco-friendly food sources that can sustainably feed the billions of people on Earth without harming the planet’s ecosystem. These new sources primarily consist of marine foods like algae, animal-based proteins like insects, and mushrooms [3,4,5]. Mushrooms, in particular, have been used for both food and medicine for thousands of years in various parts of the world and are a rich source of essential nutrients. They also possess functional properties that can benefit human health and technological applications [6].

In recent years, mushrooms have gained popularity due to their impressive nutritional profile. They are rich in protein, carbohydrates, dietary fiber, water, minerals, and vitamins, with a low fat content [7]. Mushrooms contain approximately 13–62% crude protein, 14–75% carbohydrates, and only 0.1–8% fat on a dry matter basis [8]. Furthermore, they are an excellent source of micronutrients typically found in grains, vegetables, and meats, including niacin, copper, selenium, phosphorous, and pantothenic acid. Additionally, mushrooms are the only vegetarian sources of vitamin B12 and vitamin D [9]. As a result, they can be used as a substitute for animal protein, and with the advancement of 3D printing technology in the food industry, it may be possible to create protein analogs made of edible mushrooms that closely mimic the flavor and taste of animal meat [10].

Polysaccharides are particularly important in both food production and nutrition as they provide nutritional and technological advantages. They perform various functions, such as serving as dietary fiber, energy sources, and fulfilling other detailed roles in nutrition and health. Additionally, polysaccharides are essential compounds in food production due to their technological properties, which enable them to be utilized as product stabilizers, thickening agents, water-holding substances, and more [11].

Mushrooms are rich in polysaccharides, which have been found to have numerous health benefits [12,13]. These polysaccharides, extracted from mushrooms, have also demonstrated significant antioxidant activity [14]. The yield of polysaccharides in a mushroom fruiting body varies between 4.95% and 8.21% [15]. Additionally, mushroom-extracted polysaccharides possess a range of impressive properties, such as antitumor, antiviral, antilipidemic, and immune-regulatory activities [14,16,17]. The mode of action of polysaccharides involves the gut microbiota, making them prebiotics in the digestive system [13]. They can also enhance the immune response as functional factors [18]. Furthermore, research has shown that mushroom polysaccharides can be a source of safe and effective anti-diabetic drugs with high biological activity and reasonable prices [18,19,20].

β-glucans or mixed glucans (homopolysaccharides) are the most bioactive polysaccharides found in mushrooms, accounting for 3.1–46.5% of their content. However, other types of heteropolysaccharides and polysaccharide–protein complexes, such as lectins, also possess biological properties [13,20].

*Agaricus bisporus*, commonly referred to as the white button mushroom, is one of the most widely cultivated mushroom varieties, with commercial production taking place in over 70 countries [21]. This wood-rotting fungus grows on lignocellulose substrates and is a member of the class of basidiomycetes [22]. Its popularity has risen due to its ease of cultivation, adaptability to various growing substrates, and health benefits, such as improving metabolic syndrome, gastrointestinal health, immune function, and anticancer activity [9,21]. The majority of polysaccharides found in *Agaricus bisporus* are β-glucan polymers, xylans, galactans, mannans, and chitin [13]. The β-glucan polymers in *Agaricus bisporus* consist mainly of β-(1→3) linkages with β-(1→6) branches in the main chain [23].

The most commonly used method for extracting polysaccharides from the agricultural industry is hot water extraction, which is also highly accessible and safe [24]. Furthermore, this method offers several advantages, such as having low operating costs and straightforward equipment requirements [25]. To facilitate the development of mushroom polysaccharides with a wider range of applications, it is important to consider the factors that affect the extraction efficiency and yield of polysaccharides. High-yield extracts can serve as valuable nutritional additives and technological improvers in a variety of food applications, making them a potential key ingredient.

Therefore, it is essential to address the knowledge gaps in mushroom technology, which also involves investigating mushroom polysaccharides that have been reported to possess significant nutritional and commercial potential. The potential application of mushroom polysaccharides as well as mushroom polysaccharide extracts may contribute to addressing the food and nutritional demands of society in a world where traditional food sources (plants and animal sources) are affected by climate change and environmental pollution. The richness and diversity of mushrooms may result in their broad applications for the modern food industry and nutritional demands of society; however, unless science successfully solves the problem of transferring the mycorrhiza phenomenon into mushroom cultivation, attention must be focused on cultivated species.

According to the FAOstat report in 2021, the world production of mushrooms was 42.79 million tons [26]. It is worth noting that around 100 species of mushroom can be cultivated commercially, but only a limited number of them, not exceeding 20, can be grown on an industrial scale [27]. In this group, six species, i.e., *Agaricus bisporus*, *Lentinula edodes*, *Auricularia* spp., *Pleurotus* spp., *Flammulina velutipes*, and *Volvariella volvacea* represent 90% of total production.

For the moment, these species can be taken into the consideration of larger-scale application; however, the detailed investigation of less common or wild species should also be carried out in order to understand the nutritional, technological, and other phenomena in which mushrooms might be applied.

Based on this, the objective of this investigation is to optimize the extraction of polysaccharides from *Agaricus bisporus* using a Response Surface Methodology design that defines the relationship between three factors and different response variables to identify the optimal conditions for polysaccharide extraction. Specifically, the studied factors were the water-to-mushroom ratio (W2M), extraction temperature (ET), and extraction time (EH). The results of this optimization were compared with those of the optimization performed by means of antioxidant properties measured with different methods to establish the possible mechanism of extract bioactivity.

## 2. Results and Discussion

### 2.1. Polysaccharides Concentrations and Aldehyde Group Content

Mushrooms are primarily composed of 35–70% carbohydrates in their dry matter, as per a study [28]. These carbohydrates are composed of polysaccharides, including β-glucans, mono- and disaccharides, chitin, and glycogen [29]. The presented study’s focus was on the polysaccharide content, and thus the total carbohydrate content as well as the reducing sugars content were determined.

The outcome of the experiments revealed that the amount of carbohydrates (TCC) fluctuated between 122.10 and 208.24 milligrams of glucose per gram of lyophilized *Agaricus bisporus* under the conditions of 70 °C for 4 h, 50:1 (*v*/*w*) W2M, and 90 °C for 5 h, 100:1 W2M, respectively. The DNS method allowed the establishment of the level of free aldehyde groups ranging from 0.718 to 1.312 µmol/mL of extract at 80 °C for 5 h and 150:1 W2M, and 90 °C for 4 h and 50:1 ratio, respectively.

The process of extracting polysaccharides was optimized using second-order polynomial equations, resulting in a model that demonstrated a high level of significance and a good fit with the experimental data. The model achieved an R^2^ value of 0.91 for the total carbohydrate content in the sample and 0.96 for the free aldehyde group content in the extract, as shown in Table 1.

The statistical analysis of TCC revealed a significant positive linear impact of all variables, along with negative quadratic effects. The interaction effect of X12 (time and temperature) was discovered to be significant, while the interaction effect of X13 (time and soli/liquid ratio) was found to be nonsignificant. Lastly, the interaction effect of X23 (temperature and solid/liquid ratio) was found to be significant. The ANOVA results for the examined response variable presented in Appendix A indicate that the model parameters can account for the experimental variation in the response variables.

The analysis of TCC only assesses the overall amount of carbohydrates in the extract. Although a wide range of carbohydrates are water soluble (monosaccharides, oligosaccharides, and some polysaccharides), the crucial aspect is identifying the type of carbohydrate extracted from the sample. This was the goal of the DNS analysis of the obtained extracts. The number of -CHO groups in the extracts was optimized to minimize its value, resulting in longer polysaccharide chains in solution.

In this case (Table 1), the only significant effects were detected for temperature (positive effect) and W2M (negative effect) by means of the linear coefficient. On the other hand, the only significant quadratic effect was shown for W2M. The mixed effects were detected for time and temperature (positive) and temperature solid/liquid ratio. In terms of optimal values for the extraction process (Table 2), it was found that increasing the temperature led to a higher number of –CHO groups, i.e., the lower length of polymer chains.

On the other hand, the highest TCC was noted for the solid/liquid ratio at the level of around 120. The optimal values for lowest RS were similar by means of the process time (4.78) and the liquid/solid ratio of around 130; however, the optimal temperature for this optimization was 70 °C. This is probably caused by the ability of the extraction medium, i.e., water, to cause polysaccharide chains to be hydrolyzed at a higher temperature (see Figure 1).

### 2.2. Antioxidant Activity of Agaricus bisporus Extracts

Due to their health-promoting and functional properties, natural antioxidants found in foods are becoming increasingly popular. Additionally, the development of natural antioxidants such as polysaccharides in mushrooms is expected to enhance food safety [30,31]. The study aimed to assess the antioxidant activity of carbohydrate-rich extracts of *Agaricus bisporus*, which can be influenced by various extraction parameters such as temperature, time, and liquid/solid ratio. The study focused on the effect of these parameters on the antioxidant activity of the extracts, and employed DPPH, ABTS, and H_2_O_2_ scavenging activities for the analysis. The extracts demonstrated significant scavenging activity against ABTS and hydroxyl radicals (H_2_O_2_), but did not show significant scavenging activity against DPPH radicals.

The values obtained in the DPPH analysis may be attributed to the mechanism of DPPH reduction. It is well known that DPPH can be scavenged by antioxidants that act as H-donors, while the quenching of the radical leads to the formation of a hydrazine derivative [32]. The DPPH reaction mechanism is referred to as SPLET, involving sequential proton loss and electron transfer. This mechanism was also observed for ABTS, but in this case, the optimal environment for the reaction was water, where DPPH alcohol (MeOH or EtOH) or alcohol/water mixtures were preferred for scavenging [33]. During the pretreatment stage involving washing the lyophilized mushroom with MeOH, any low-molecular weight compounds that were alcohol soluble were washed away. Furthermore, the low content of reducing sugars indicates that the carbohydrates in the extracts have a higher molecular mass, making them poorly soluble in MeOH. As a result, the percentage of scavenged radicals ranged from 0.19% to 3.45% at 70 °C and 4 h, and 90 °C and 4 h, respectively, with a high standard deviation observed. It is also noteworthy that the optimization procedure for DPPH scavenging was unsuccessful, as evidenced by the R^2^ value being below 0.70.

The outcomes obtained with the H_2_O_2_ technique varied between 54.99% and 82.41% at temperatures of 70 °C, time durations of 4 h, 50 mL/g W2M and 90 °C, 4 h, 150 mL/g W2M, respectively. This indicates the high efficacy of the extracts in neutralizing hydroxyl radicals. However, in this instance, the fitting of the model using the suggested methodology was also unsuccessful, as observed in the case of DPPH (R^2^ < 0.45). This may be attributed to the high reactivity of the OH radicals (average lifetime of ~10^−9^ s) [34]. The high reactivity in a dense aquatic environment (macromolecular solution) can cause the OH radical to be quenched by colliding with macromolecules rather than through typical scavenging mechanisms. This phenomenon may be responsible for the random values of scavenging and the lack of success in optimization.

In contrast, the ABTS scavenging activity demonstrated a range of 45.35% to 91.57% at 70 °C for 4 h and 150 mL/g W2M, and 90 °C for 4 h and 150 mL/g, respectively. The optimization of ABTS was successful, as evidenced by an R^2^ value greater than 0.93, which also provides information about the composition of the sample and scavenging activity. The antioxidant action of polysaccharides is attributed to few mechanisms, which include polysaccharide conjugates (e.g., phenols), antioxidant active impurities in the polysaccharide matrix, the chelation of metal ions, and the presence of specific functional groups that are introduced by means of chain modification or natural and structural features of the chain [35,36]. In the present study, none of these mechanisms can be definitively ruled out using the applied methodology, although the less probable mechanism is polysaccharide conjugates and impurities in the polymer matrix due to the low DPPH scavenging values. The high results obtained in the case of ABTS and H_2_O_2_ may be attributed to chelating or the presence of functional groups; however, further detailed mechanistic studies are required to confirm this phenomenon.

The performed optimization of the influence of extraction parameters on antioxidant activity shows that the three linear factors influence the ABTS radical scavenging activity as shown in Table 3. It was demonstrated that temperature, time, and liquid/solid ratio have a significant effect on antioxidant activity; however, only in case of liquid/solid ratio is the parameter negative. On the other hand, the temperature with liquid ratio relation is also statistically significant and negative. According to the response surface (Figure 2), when making a relation between temperature, time, and ABTS radical scavenging activity, it can be demonstrated that the antioxidant activity of the extract is significantly increased to more than 80% when increasing the temperature and time up to 5 h. Additionally, the calculated optimal values for the maximization of the antioxidant action of the extract (Table 4) shows that, as in other discussed cases, the optimal time for the highest antioxidant scavenging activity is near the maximum time investigated, i.e., around 5 h. Contrarily, the strongest antioxidant action is observed at a temperature around 80 °C and with a low liquid-to-solid ratio (around 50:1 mL/g).

### 2.3. Validation of Optimized Models

The optimal conditions were determined using Statistica 13.1 (StatSoft, Poland) by maximizing/minimizing the desirability of the responses. The desirability was maximized at the highest concentration of TCC and the highest values for ABTS, and the RS parameter was minimized. These optimal conditions were used for the extraction process, and the responses were determined and validated according to a predetermined procedure. The obtained desirability was 0.897 for TCC, 0.888 for RS, and 0.967 for ABTS. Under these optimal conditions, the experimental values were in agreement with the predicted values, with the coefficient of variation (CV) ranging from 6 to 10, as shown in Table 5. The coefficient of variation provides a measure of the dispersion of data.

### 2.4. Summary

The majority of the available literature data on the extraction of polysaccharide fractions from *Agaricus bisporus* focus on enzymatic methods, alkali acid extraction, and the utilization of microwaves or ultrasound [26,37,38]. The hot water extraction (HWE) method is less frequently employed, and the available data indicate that fractions can be obtained using this method with yields unable to exceed 25 mg/g (purified fraction) [39]. In contrast to other methods, hot water extraction is the most convenient method and is widely utilized in laboratories and industry [40]. Through the application of HWE, the plasma walls of the cells may be easily separated, which results in the dissolution of the vacuoles contents in an external solvent, i.e., water [41]. Furthermore, the HWE method possesses an advantage in that extracts can be utilized directly for food production without additional purification. A disadvantage of the HWE method discussed in the literature is the potential for polysaccharide degradation, which affects extraction efficiency [37]. Additionally, while more sophisticated extraction methods, such as enzymatic (EAE), alkali/acid extraction (AAE), or using microwaves or ultrasounds, may be employed to obtain polysaccharide fractions from mushroom samples, the direct application of these extracts or purified fractions may present challenges, including the necessity to remove side products (enzymes, salts, etc.), high energy input, or complex procedures/equipment.

In the present study, the negative effect of temperature on the size of the macromolecules in the solution, manifested by an increase in the quantity of reducing sugars in the solution, was confirmed.

The antioxidant properties of polysaccharides from *Agaricus bisporus* have been determined in several studies to date; however, these investigations have primarily involved extracts/preparations obtained via enzymatic extraction or alkali acid extraction (AAE) methods [42,43]. Due to the differences in the molecular structures of the fractions obtained using these methods, it is not feasible to directly compare them with the fractions obtained using HWE. Nevertheless, given the results obtained, the polysaccharide fraction of HWE also exhibited antioxidant properties. However, considering the test methods employed, the mechanism of free radical scavenging is, in this case, distinct from that observed for the enzymatic-assisted extraction (EAE) and AAE extracts.

## 3. Materials and Methods

### 3.1. Materials

*Agaricus bisporus* was acquired from a local market in Krakow, Poland, as fresh fruiting bodies. The chemical reagents, such as DPPH (2,2-Diphenyl-1-picrylhydrazyl), ABTS (2,2′-azino-bis(3-ethylbenzothiazoline-6-sulfonic acid)), methanol, persulfate, K_2_HPO_4_, KH_2_PO_4_, aluminum chloride, potassium acetate, phenol, H_2_SO_4_, DNS (3,5-dinitrosalicylic acid) reagent, D-glucose, NaOH, and potassium sodium tartrate, were sourced from Sigma-Aldrich (Poznan, Poland) and utilized as pure for analysis.

### 3.2. Methods

#### 3.2.1. Variables and the Extraction Process

The extraction process was investigated using the following variables: time of extraction ranging from 3 to 5 h (X1), temperatures from 70 to 90 °C (X2), and liquid-to-solid ratio from 50:1 to 150:1 *v*/*w*, mL/g (X3). These values were selected based on preliminary studies. The following dependent variables were measured: total carbohydrate content (TCC, mg/g), the amount of free aldehyde groups (reducing sugars—RS, µmol/mL) determined using DNS, the % of radical scavenging ability assessed using DPPH, H_2_O_2_ (hydroxyl radical scavenging), and ABTS.

The extraction procedure comprised several steps. Fresh mushrooms were carefully cleaned with cold tap water and left to dry at room temperature for 5 h. After drying, the mushrooms were immediately frozen to −18 °C and then freeze-dried using a benchtop freeze dry system (FreeZone 6 Liter Benchtop Freeze Dry System; Labconco, Kansas City, MO, USA). Next, the sample was grinded for 5 min using a laboratory grinder. To remove low-molecular compounds, including simple carbohydrates, the pretreatment involving stirring (100 rpm) the samples with pure methanol (solid–liquid ratio 1:100 *w*/*v*) at room temperature for 2 h was applied. Following pretreatment, the samples were filtered through a paper filter and solid pretreated samples were left to dry for 24 h at room temperature [25].

Next a 1.00 g of lyophilized pretreated sample was mixed with different quantities of deionized water (X3) under different temperatures (X2) and times (X1) with constant stirring according to Table 1. After finishing the extraction, the sample was centrifuged at 9000 rpm for 10 min using a high-speed brushless centrifuge (MPW-350R, MPW, Warszawa, Poland) and the supernatant was collected and stored at −18 °C for analysis [44]. Each extraction assay (run) was performed six times.

#### 3.2.2. Experimental Design

The optimization of the experiment was accomplished through the use of Response Surface Methodology (RSM) for the extraction of polysaccharides (TCC), the reducing sugars content, and antioxidant activity (DPPH, ABTS, and H_2_O_2_). A Box–Behnken experiment design was employed to establish a second-order polynomial model for the extraction of polysaccharides in relation to their antioxidant properties. Five responses were studied, including the following: the total carbohydrate content (TCC), the level of reducing aldehyde groups as a measure of the average chain length of polysaccharides, and antioxidant activity through radical scavenging by DPPH, ABTS, and H_2_O_2_. All responses were evaluated based on the combined effect of three parameters, including extraction time (X1), temperature (X2), and solvent-to-solid ratio (X3), each studied at three levels corresponding to low, medium, and high levels, respectively. The Box–Behnken matrix consisted of 15 experiments, including triplicates at the central point. The variables and experimental design levels used in the Box–Behnken design are presented in Table 6.

This applied approach allows for the assessment of the effects of the factors and the optimization of the impact of the independent variables on the extraction process. By employing multiple regression analysis on the experimental data, the optimized values of extraction time, process temperature, and solvent-to-solid ratio were predicted [12]. A second-order polynomial model equation was applied to represent the generalized mathematical quadratic response surface. The equation for this model is given as follows [45]:(1)$$y=\beta_{0}+\overset{k}{\underset{i=1}{\sum }}\beta_{i}x_{i}+\left(\overset{k-1}{\underset{i=1}{\sum }}\overset{k}{\underset{i=j=1}{\sum }}\beta_{ji}x_{j}x_{i}\right)+\overset{k}{\underset{i=1}{\sum }}\beta_{ii}x_{i}^{2}+\varepsilon$$
where

*y* is the response value;

*k* is the number of variables. For the three independent variables investigated in this experimental design, *k* = 3;

*β*_0_ is defined as the constant effect;

*β_i_* is the linear regression coefficient;

*β_ii_* is the quadratic regression coefficient;

*β_ji_* is the interaction regression between the parameters;

*x_i_* and *x_j_* are represented by the levels of the independent coded variables;

*ε* is the error.

In this study, the Statistica 13.1 software (Statsoft, Kraków, Poland) was employed to analyze the response surface.

To assess the impact of each factor (temperature, time, and ratio) on the response variable, an analysis of variance (ANOVA) was conducted with a 95% confidence level. The regression coefficient (R^2^), the *p*-value of the regression model, and the *p*-value of the lack of fit (LOF) were analyzed to determine the accuracy of the regression model (Appendix A). Optimal conditions were calculated using the Statistica 13.1 software (Statsoft, Kraków, Poland). Optimal conditions were validated based on the maximum total carbohydrate content (TCC), the amount of reducing aldehyde groups in the extract, and antioxidant activities (DPPH, ABTS, and H_2_O_2_), as determined using RSM. The experimental values were compared with the predicted values based on the coefficient of variation (CV%) to validate the predictive model. The validity and adequacy of the predictive extraction (*n* = 3) were verified using a two-sided *t*-test (*p* = 0.05) to compare the predictions with the observed values.

#### 3.2.3. Total Carbohydrate Content Analysis Using Colorimetric Dubois Method

The Dubois method was applied for the TCC content analysis [39,46]. Specifically, 1.0 mL of 5% phenol solution was combined in a test tube with 1.0 mL of sample extract, followed by the addition of 5.0 mL of concentrated sulfuric acid (96%). The test tubes were then cooled to room temperature and mixed. Colorimetric analysis was carried out using a UV-Vis spectrophotometer (UV-VIS Dual Beam UVS-2800, Labomed Inc., Los Angeles, CA, USA) at a wavelength of 490 nanometers, in comparison to a blank (water + reagents) prepared under identical conditions. A glucose solution was employed to create a standard curve, and the results were reported in milligrams of glucose. All samples were analyzed in triplicate.

#### 3.2.4. Quantification of Free Aldehyde Group—Reducing Sugars

3,5-dinitrosalicylic acid (DNS) is an indicator of the presence of a free aldehyde group, as it can be easily reduced to 3,5-diaminosalicylic acid (DNSA). The amount of DNSA can be easily determined using a spectrophotometric method. The amount of DNSA in the sample is correlated with the number of free aldehyde groups, that is, the number of reducing ends of carbohydrates. As a result, it is simple to determine whether the carbohydrates in the extract are polymeric or oligomeric. The number of -CHO groups in the extracts was optimized to minimize its value, resulting in longer polysaccharide chains in the solution. The applied DNS analysis was carried out as follows [47]. A 1.0 mL of the extract was combined with 0.75 mL of DNS solution and 3.25 mL of distilled water. The mixture was heated at 80 °C in a water bath for 10 min, followed by cooling in water for 20 min. The reaction was examined using the UV-Vis technique (Labomed Inc., USA) at a wavelength of 540 nm. A blank sample was prepared in the same manner, using water instead of the sample. A glucose solution was utilized to formulate the standard curve, and the results were reported as ppm of glucose. All samples were analyzed in triplicate.

#### 3.2.5. DPPH Radical Scavenging Activity

To evaluate the potential of the extract to scavenge DPPH radicals, a standard procedure was employed [48]. A 0.1 mL aliquot of the extract was added to a 3.9 mL solution of DPPH (prepared by adding 22 mg of DPPH to 50 mL of methanol and then diluting with methanol to an absorbance of 0.8 ± 0.02 at 515 nm). The mixture was vigorously vortexed for 30 s prior to being allowed to sit in the dark at room temperature for 30 min. The decolorization of DPPH was measured using a UV-Vis spectrophotometer (Labomed Inc., USA) at 515 nm, and the DPPH free radical scavenging activity (%) was calculated as the percentage decrease in absorbance. The blank sample was prepared with the methanolic dilution of DPPH (0.1 mL of pure methanol was added to a 3.9 mL solution of DPPH).

#### 3.2.6. ABTS Radical-Scavenging Activity

ABTS+ solution was prepared by mixing an equal volume of 7 mM ABTS and 2.4 mM potassium persulfate and allowing the solution to incubate for 12–16 h at room temperature in the dark. Afterward, the solution was diluted with distilled water to achieve an initial absorbance of 0.70 ± 0.02 at 734 nm. The reaction was initiated by adding 50 µL of the sample to 950 µL of water, followed by the addition of 1 mL of ABTS+ solution. The mixture was then incubated for 7 min at room temperature. The decrease in absorbance was measured at 734 nm, and the ABTS+ scavenging effect was calculated as a percentage of the decrease in absorbance. An equal amount of ABTS and water were used as control, respectively [49].

#### 3.2.7. Hydroxyl Radical Scavenging Activity Assay

The hydroxyl radical scavenging activity was determined by mixing a 0.6 mL sample with 1.8 mL of 50 mM phosphate buffer (pH = 7.4) and 3.6 mL of H_2_O_2_ solution (2 mM). The reaction was vortexed and held for 10 min at room temperature. The absorbance decrease was measured at 230 nm, and the scavenging activity (%) was calculated as a percentage of the absorbance reduction. The blank sample was prepared by mixing 0.6 mL of distilled water with a 3.6 mL working solution, i.e., H_2_O_2_ in buffer [48].

#### 3.2.8. Model Validation

The optimal conditions for polysaccharide extraction (including all variables: EH, ET, and W2M) were tested for the highest carbohydrate content while minimizing the free aldehyde group content and maximizing antioxidant activity (as measured using ABTS and hydroxyl radical scavenging). These conditions were determined using RSM, and the resulting experimental values were compared to those predicted by the model to ensure their accuracy. All responses were re-measured under the optimized extraction conditions to confirm the results.

## 4. Conclusions

The *Agaricus bisporus* mushroom is a significant source of polysaccharide macromolecules that have potent antioxidant properties. These macromolecules can be extracted using solid-to-hot water extraction, which is a cost-effective and environmentally friendly method compared to the use of organic solvents. The most efficient conditions for extracting the polysaccharide-rich fraction were found to be 5 h at 90 °C with a liquid-to-solid ratio of 118.59 mL/g. On the other hand, to obtain the longest chain (minimize the amount of reduced sugar fraction), the process conditions of 4.78 h, 70.45 °C, and a liquid-to-solid ratio of 128.82 mL/g were optimal. Finally, the antioxidant power of the extract was maximized at a process run of 4.96 h, a temperature of 82.39 °C, and a ratio of 51.0:1 mL/g. Further research is warranted to investigate the mechanisms of the antioxidant activity of the extracted polysaccharides and to determine the detailed molecular structure of the polysaccharides extracted from mushroom tissue.

## Figures and Tables

**Figure 1 molecules-29-04783-f001:**
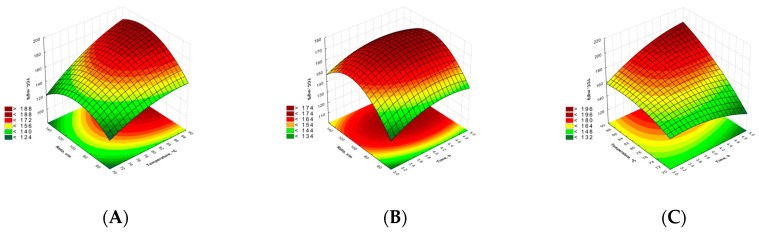
Response surface plots for TCC (**A**–**C**) and RS (**D**–**F**).

**Figure 2 molecules-29-04783-f002:**
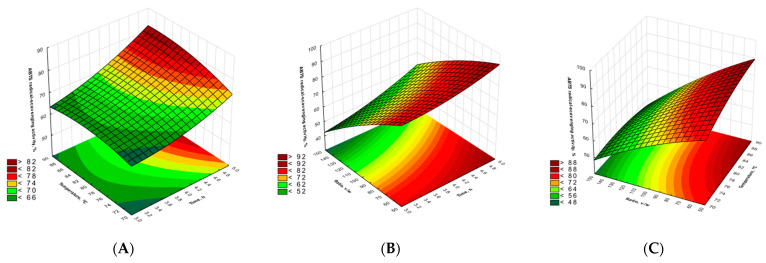
Response surface plots for ABTS antioxidant scavenging (**A**–**C**).

**Table 1 molecules-29-04783-t001:** Regression coefficient (β) and coefficient of determination (R^2^) values of the predicted models for TCC and RS (reducing sugar content).

	TCC	RS
Intercept		
β_0_	172.181	0.824
Linear		
β_1_	11.782	*p* > 0.01
β_2_	44.536	0.093
β_3_	17.613	−0.465
Quadratic		
β_11_	−16.371	*p* > 0.01
β_22_	−15.496	*p* > 0.01
β_33_	−43.625	0.329
Mixed		
β_12_	30.164	0.0699
β_13_	*p* > 0.01	*p* > 0.01
β_23_	17.128	−0.083
R^2^	0.915	0.957

**Table 2 molecules-29-04783-t002:** Optimal value for TCC content (maximal values in investigated range) and RS (minimal value).

		Predicted Response				
	TCC	Maximal, mg/g	±95% CI	RS	Minimal, µmol/mL	±95% CI
Time, h	5.0	202.51	196.38–207.91	4.78	0.68	0.658–0.717
Temperature, °C	90.0			70.45		
Ratio, *w*/*v*	118.59			128.82		

**Table 3 molecules-29-04783-t003:** Regression coefficient (β) and coefficient of determination (R^2^) values of the predicted models for DPPH, ABTS, and hydroxyl radicals scavenging.

	DPPH	ABTS	H_2_O_2_
Intercept			
β_0_	*p* > 0.01	70.303	74.716
Linear			
β_1_	−0.926	14.773	*p* > 0.01
β_2_	−1.888	4.949	*p* > 0.01
β_3_	0.765	−36.289	9.449
Quadratic			
β_11_	1.628	*p* > 0.01	*p* > 0.01
β_22_	2.159	*p* > 0.01	*p* > 0.01
β_33_	*p* > 0.01	*p* > 0.01	*p* > 0.01
Mixed			
β_12_	*p* > 0.01	*p* > 0.01	12.927
β_13_	−1.144	*p* > 0.01	*p* > 0.01
β_23_	*p* > 0.01	−7.045	*p* > 0.01
R^2^	0.661	0.937	0.41

**Table 4 molecules-29-04783-t004:** Optimal value for ABT scavenging activity.

Predicted Response			
	ABTS	Maximal, %	±95% CI
Time, h	4.96	94.69	91.82–97.57
Temperature, °C	82.39		
Ratio, *w*/*v*	51.01		

**Table 5 molecules-29-04783-t005:** The validation of predicted values using experimental data at optimal extraction conditions.

Dependent Variables	Predicted Value	Experimental Value	%CV
TCC, mg/g	202.51	209.98	8.79
RS, µmol/mL	0.68	0.61	9.68
ABTS, %	94.69	95.99	6.01

**Table 6 molecules-29-04783-t006:** Experimental factors and measured values of responses.

Run	Independent Variables			TCC, mg/g	-CHO, µmol/mL	DPPH	ABTS	H_2_O_2_
	Time (X1), h	Temperature (X2), °C	Ratio *w*/*v* (X3)			%		
1	3	70	100	134.42 ± 2.58	0.814 ± 0.018	3.45 ± 0.23	61.52 ± 1.47	77.82 ± 6.96
2	3	80	50	133.10 ± 3.88	1.236 ± 0.078	0.25 ± 0.19	86.72 ± 1.69	68.11 ± 2.08
3	3	80	150	153.37 ± 4.40	0.736 ± 0.047	3.18 ± 0.99	45.35 ± 0.76	74.32 ± 3.32
4	3	90	100	152.40 ± 9.51	0.808 ± 0.039	2.23 ± 0.99	57.58 ± 3.53	61.90 ± 10.98
5	4	70	50	122.10 ± 2.83	1.106 ± 0.035	2.39 ± 0.56	74.60 ± 3.13	54.99 ± 8.47
6	4	70	150	133.07 ± 2.41	0.754 ± 0.037	3.04 ± 0.97	46.21 ± 2.83	74.45 ± 2.65
7	4	80	100	172.18 ± 3.73	0.824 ± 0.053	0.52 ± 0.40	70.30 ± 2.69	74.72 ± 1.08
8	4	80	100	174.41 ± 3.61	0.870 ± 0.007	0.54 ± 0.44	67.88 ± 0.30	75.07 ± 1.31
9	4	80	100	169.96 ± 2.62	0.779 ± 0.031	0.50 ± 0.45	72.73 ± 0.61	74.37 ± 0.92
10	4	90	50	145.90 ± 2.41	1.312 ± 0.063	0.19 ± 0.07	91.57 ± 0.95	68.33 ± 3.57
11	4	90	150	180.27 ± 3.77	0.794 ± 0.023	0.31 ± 0.12	49.09 ± 2.71	82.41 ± 6.71
12	5	70	100	129.93 ± 5.40	0.746 ± 0.034	2.68 ± 0.51	78.99 ± 2.24	67.76 ± 5.17
13	5	80	50	133.29 ± 1.91	1.207 ± 0.021	0.40 ± 0.04	90.66 ± 2.75	77.17 ± 2.48
14	5	80	150	148.97 ± 8.79	0.718 ± 0.025	1.04 ± 0.31	57.73 ± 4.82	75.20 ± 1.64
15	5	90	100	208.24 ± 3.35	0.880 ± 0.015	1.48 ± 0.04	82.88 ± 1.73	77.69 ± 0.60

values are reported as mean ± SD (*n* = 6).

## Data Availability

The data are included in this article.

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
