# Peer review of "The Optimization of the Hot Water Extraction of the Polysaccharide-Rich Fraction from Agaricus bisporus"

_molecules, 2024, doi:10.3390/molecules29194783_

Round 1

Reviewer 1 Report

Comments and Suggestions for Authors

The experimental article "Optimisation of hot water extraction of the polysaccharide-rich fraction from Agaricus bisporus" describes a study aimed at isolating a polysaccharide from Agaricus bisporus mushrooms. The introduction is very well described, from which the hypothesis of the scientific study is clear. A large volume of research has been carried out. However, there are the following comments:

1. 257-268. The description of the methods should be transferred to the materials and methods.

2. Compare the data obtained in this work with the data presented in the literature. Conclude how promising it is to use Agaricus bisporus to isolate polysaccharides with antioxidant activity.

Reviewer 2 Report

Comments and Suggestions for Authors

The authors carefully studied the extraction of polysaccharide in hot water by drying, grinding and stirring with solvent to remove impurities from the mushroom bisporus as raw material,

and clearly verified the antioxidant effect of the extract with three types of free radicals. However, I don't think this article is very innovative; First of all, polysaccharide extraction technology has been studied in repeat times, and response surface is also a common optimization method. DPPH radical scavenging activity, ABTS radical-scavenging activity, Hydroxyl radical scavenging activity assay are also routine method of detection.

This research has some practical significance. However the research method is very traditional, and the author has many writing formats that do not meet the requirements of scientific papers, such as:

Line 112 what does single % mean

 ml should replaced by mL pleast check the whole manuscript

Line 121what are the temperature and speed of strring ?  is the solid-liquid ratio of 1:100

Line 122 Following pretreatment, the samples were filtered through a paper filter...  Filter for what? Methanol extract?

Please use a regular three-wire table for the table

And the page number

Reviewer 3 Report

Comments and Suggestions for Authors

The manuscript entitled "Optimisation of hot water extraction of the polysaccharide rich fraction from Agaricus bisporus" by Khalil et al., studied the extraction parameters of polysaccharides from Agaricus bisporus mushroom using solid-to-hot water extraction. The authors reported the antioxidant properties of the extracted polysaccharide macromolecules. 

Specific Concerns: 

1. A summary of other candidates that can be utilized in the extraction of polysaccharides and other bio-active components by the current method would be useful.

2. What are the significant advantages of Agaricus bisporus over the other species?

3. Include a table summarizing all the different homopolysaccharides and heteropolysaccharides extracted and purified by the current method.

4. A detailed description about the functional benefits of the different polysaccharides extracted by the current method would be helpful.

5. Describe about the controls used in experiments mentioned in the methods section form 2.2.5 to 2.2.7.

6. What are other bio-active molecules that are present in the extract?

7. Describe about the potential benefits of all the other potential candidate bio-active molecules that are co-extracted by the current method?

8. Can such biomolecules interfere or influence the antioxidant activity of the extract?

9. Does the polysaccharides extracted by current method have any effect in the inhibition of lipid peroxidation.

10. Does Agaricus bisporus extracted polysaccharides effects the activity of any antioxidant enzymes such as catalase or superoxide dismutase etc.?

11. Elaborate on the advantages and limitations of the current method compared to the other reported procedures. 

Reviewer 4 Report

Comments and Suggestions for Authors

The manuscript describes the optimization of hot water extraction of the polysaccharide-rich fraction from Agaricus bisporus. The authors conducted a study to optimize the extraction parameters including process time, temperature, and liquid-to-solid ratio to obtain the polysaccharide-rich fraction from lyophilized Agaricus bisporus. The extracts obtained were then analyzed for total carbohydrate content, reducing sugars, and antioxidant activity using DPPH, ABTS, and hydroxyl radical scavenging assays. The authors found that extraction time did not significantly affect polysaccharide yield, but noted significant effects from other parameters. They discovered that the highest carbohydrate content was achieved with a longer process time, higher temperature, and a liquid-to-solid ratio of 118 mL/g. Additionally, they found that lower temperatures reduced sugar levels, while optimal antioxidant activity required moderate temperatures and low liquid-to-solid ratios. In conclusion, the study suggests that the aqueous extraction method is effective in recovering bioactive polysaccharide fractions from Agaricus bisporus, which also exhibit promising antioxidant properties. I wo recommend publishing this manuscript in Molecules journal.

Round 2

Reviewer 2 Report

Comments and Suggestions for Authors

I always think that there is not enough innovation to support the publication of articles.

Reviewer 3 Report

Comments and Suggestions for Authors

The manuscript entitled "Optimization of hot water extraction of the polysaccharide rich fraction from Agaricus bisporus" by Khalil et al., had been revised accordingly. 

Comments on the Quality of English Language

English language is alright.